# Review of Laboratory Testing and Biomarker Screening for Preeclampsia

Antonia F. Oladipo [1],* and Maansi Jayade [2]

1   Department of Obstetrics and Gynecology, Hackensack University Medical Center, Hackensack, NJ 07601, USA
2   Hackensack Meridian School of Medicine, Nutley, NJ 07110, USA; maansi.jayade@hmhn.org
*   Correspondence: antonia.oladipo@hmhn.org

**Abstract:** The purpose of this review is to elucidate the different laboratory and biomarker testing methods available for screening and diagnosis of preeclampsia. These include routine testing, such as blood pressure readings, qualitative and quantitative urine testing, complete blood count with platelets, serum creatinine levels, liver chemistries, and serum bilirubin levels. This review also details the use of non-routine testing, such as screening for angiogenic and anti-angiogenic markers, such as placental growth factor (PlGF) and soluble fms-like tyrosine kinase-1 (sFlt-1). Blood pressure measurements and proteinuria are the most routinely used screening tools used for preeclampsia and there are limited data on the utility of other screening techniques because of a greater focus on the etiology and treatment of preeclampsia. Similarly, serum angiogenic biomarkers are not routinely collected, so there is limited evidence regarding using them as screening tools for preeclampsia and more data are needed to determine their significance in the screening and diagnosis of preeclampsia.

**Keywords:** preeclampsia; proteinuria; blood pressure measurement; serum markers; pro-angiogenic biomarkers; anti-angiogenic biomarkers; preeclampsia clinical features; hypertensive disorders of pregnancy





## 1. Introduction

Preeclampsia is a multisystem disease defined as new-onset hypertension after 20 weeks of pregnancy with proteinuria, or new-onset hypertension after 20 weeks of pregnancy with no proteinuria and signs of end-organ dysfunction [1]. The American College of Gynecology (ACOG) defines hypertension in pregnancy as a systolic blood pressure $\geq$ 140 mmHg or diastolic blood pressure $\geq$ 90 mmHg that occurs at least twice, 4 h apart, after 20 weeks of gestation. Severe hypertension is defined as a systolic blood pressure $\geq$ 160 mmHg or diastolic blood pressure $\geq$ 110 mmHg [2]. Evidence of end-organ dysfunction includes signs of renal insufficiency, impaired liver function, pulmonary edema, thrombocytopenia, and cerebral or visual symptoms. This pregnancy-related disorder usually resolves in the days or weeks after delivery [3,4].

Preeclampsia can be further subdivided into early-onset and late-onset. Early-onset preeclampsia develops prior to 34 weeks of gestation and late-onset develops at or after 34 weeks of gestation. Early-onset preeclampsia is considered a fetal disorder and is associated with placental dysfunction, intrauterine growth restriction, perinatal death, and poor maternal and neonatal outcomes. On the other hand, late-onset preeclampsia is more of a maternal disorder and is associated with a normal placenta, normal fetal growth, and more favorable maternal and neonatal outcomes [5]. While there are limited studies comparing maternal and neonatal outcomes between early-onset and late-onset preeclampsia, preliminary studies have found more adverse outcomes in patients with early-onset preeclampsia than late-onset, but this difference was not found to be statistically significant [6].

Routine screening for preeclampsia consists of evaluating for signs of end-organ dysfunction through routine blood pressure measurements and urine protein assessment. Serum biomarkers whose imbalance has been associated with preeclampsia are not yet routinely used in preeclampsia screening. These biomarkers include angiogenic factors, such as anti-angiogenic soluble fms-related tyrosine kinase 1 (sFlt-1), and pro-angiogenic factors, like placental growth factor (PlGF) and vascular endothelial growth factor (VEGF) [7,8]. These angiogenic biomarkers show promising results but further research is needed to determine the value in screening and management for preeclampsia [9,10].

## 2. Methods

There are 99 articles used in this review, the majority of which were chosen from the PubMed database. The keywords used in the search were: preeclampsia, proteinuria, blood pressure measurement, serum markers, pro-angiogenic biomarkers, anti-angiogenic biomarkers, preeclampsia clinical features, and hypertensive disorders of pregnancy. The articles were chosen based on relevancy to the review topic and date of publication. The search was carried out in January 2024, with supplementary searches performed in April 2024.

## 3. Etiology and Epidemiology

Preeclampsia accounts for 2–10% of pregnancies worldwide and is the cause of 15–20% of all preterm births. A review from the World Health Organization states that hypertensive disorders in developed countries account for 16% of maternal deaths. Risk factors for preeclampsia include nulliparity, multiple fetal gestation, hydatidiform mole, advanced maternal age, higher BMI, pregestational diabetes, renal disease, family history of preeclampsia, maternal lupus, antiphospholipid antibody syndrome, and membership in minority ethnic or otherwise disadvantaged groups [11–13]. The etiology of preeclampsia is largely unknown but is thought to involve placental insufficiency, endothelial dysfunction, and angiogenic imbalance [14,15].

## 4. Pathophysiology

The pathophysiology of preeclampsia is not fully understood, but has been thought to involve the following pathways: abnormal placental implantation, vascular endothelial damage and oxidative stress, anti- and pro-angiogenic factors, genetics, and immunologic factors [16].

Abnormal placental implantation can occur via endovascular invasion, due to deficits in the differentiation of cytotrophoblasts, causing the formation of narrow spiral arterioles, resulting in placental hypoperfusion, insufficiency, and hypoxic trophoblast tissue [14,16].

There are many studies showing a bidirectional causal relationship between hypoperfusion of the placenta and its abnormal development. Hypoperfusion, ischemia, and hypoxia are thought to be responsible for the production of factors such as sFlt-1, VEGF, and PlGF that cause maternal vascular injury and inflammation, leading to maternal hypertension and proteinuria, which are among the hallmark symptoms of preeclampsia [17,18].

Anti-angiogenic factors, such as sFlt-1, and pro-angiogenic factors, like PlGF and VEGF, are measured in relation to each other and can correlate to the onset and severity of preeclampsia [7]. PlGF is predominantly expressed in the placenta and has a major role in the development and maturation of the placental vascular system. PlGF supports trophoblast growth, and low levels of PlGF are a marker for abnormal placentation, which is seen in preeclampsia [19,20]. Figure 1 shows that patients with preeclampsia have lower levels of circulating PlGF concentrations than those without preeclampsia. Chau et al. also visually demonstrated the various stages of utero-placental maturation corresponding to different levels of PlGF. Levels of PlGF peak in the second trimester and may play a role in trophoblast proliferation [21].

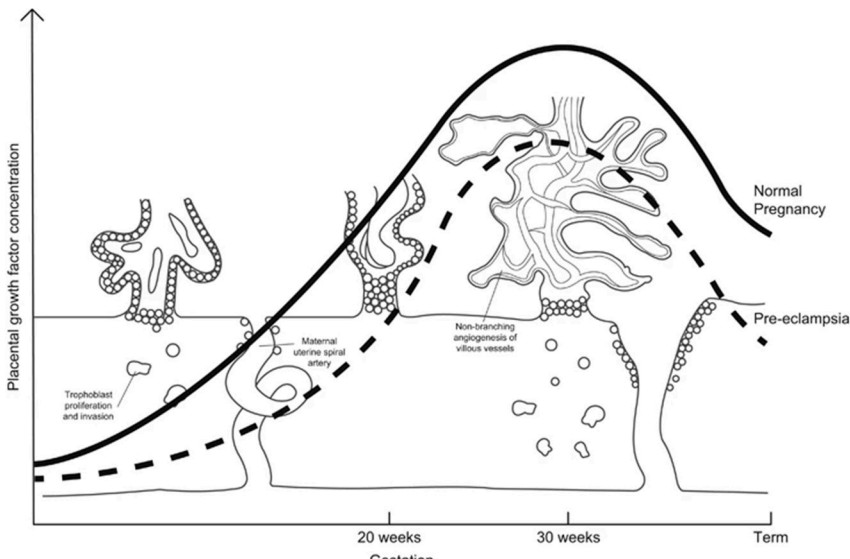

**Figure 1.** Graphical representation of levels of circulating placental growth factor (PlGF) plotted against weeks of gestation. The solid line represents levels of PlGF in a non-preeclamptic pregnancy and the dashed line represents those levels in preeclampsia. Levels of circulating PlGF peak in the second trimester, which coincides with maturation of the utero-placental circulation, suggesting that PlGF contributes to trophoblast proliferation [21].

The anti-angiogenic factor, sFlt-1, binds to PlGF and VEGF in preeclampsia and results in vasoconstriction and endothelial dysfunction [21,22]. A high sFlt-1/PlGF ratio may be associated with the development of preeclampsia and adverse pregnancy outcomes. VEGF, a proangiogenic factor, promotes the stabilization of endothelial cells in blood vessels, the liver, brain, and kidneys, all organs affected by preeclampsia. Animal studies show that in models with increased anti-VEGF activity, there is more renal impairment and subsequent proteinuria. Table 1 summarizes the levels of sFlt-1, PlGF, and VEGF in preeclampsia.

**Table 1.** This table details the anti- and pro-angiogenic factors that are upregulated or downregulated in preeclampsia.

| Vasoactive Factor | Type of Angiogenic Factor | Level of Factor in Preeclampsia | Source of Factor | Effects | References |
|---|---|---|---|---|---|
| Soluble fms-like tyrosine kinase-1 (sFlt-1) | Anti-angiogenic factor | Up | Placenta | Binds to PlGF and VEGF and results in vasoconstriction and endothelial dysfunction | [21] |
| Placental growth factor (PlGF) | Pro-angiogenic factor | Down | Placenta | Development and maturation of the placental vascular system | [19,20] |
| Vascular endothelial growth factor (VEGF) | Pro-angiogenic factor | Down | Placenta | Promotes the stabilization of endothelial cells in blood vessels, the liver, brain, and kidneys | [23,24] |

The factor sFlt-1 binds to VEGF, and preeclampsia is postulated to occur when the functional activity of sFlt-1 surpasses that of VEGF. Figure 2 depicts a visual representation of the relationship between sFlt-1, PlGF, and VEGF in pregnancies with and without preeclampsia [25]. Increased levels of sFlt-1 during preeclampsia inhibit VEGF and PlGF signaling and cause endothelial dysfunction [23–25]. Renal capillary endothelium is partic-

ularly sensitive to the activity of VEGF, and a sFlt-1 dependent deficiency of VEGF could be the reason why proteinuria, a sign of renal impairment, occurs in preeclampsia [24,26].

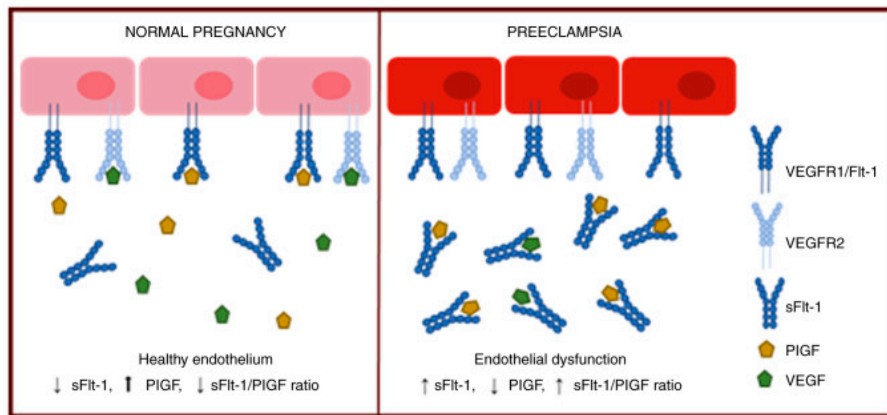

**Figure 2.** This schematic depicts the interactions between transmembrane proteins and growth factors during non-preeclamptic and preeclamptic pregnancies. In preeclampsia, excess secretion of sFlt-1 from the placenta inhibits VEGF and PlGF signaling and contributes to endothelial dysfunction [25].

Genetically inherited thrombophilias, such as Factor V Leiden, antithrombin (previously called antithrombin III), protein C, and protein S, are also hypothesized to have a role in the preeclampsia pathogenesis due to clot formation in the placenta leading to placental insufficiency, although the literature is divided. The main cause of the controversy regarding thrombophilias associated with preeclampsia is the lack of consistent prospective studies and the reliance on retrospective studies that carry a bias towards the group with preeclampsia as opposed to the control group [27,28].

A genome-wide association study (GWAS) in 2023 found 18 loci that were highly associated with preeclampsia, eclampsia, and/or gestational hypertension. These 18 loci were involved in angiogenesis and endothelial function, natriuretic peptide signaling, renal glomerular function, and immune dysregulation [29]. There are additional genes, related to immune function and dysregulation, that have been discovered which are associated with severe pregnancy hypertensive disorders. These genes include both maternal and fetal sources and include genes that have been known to be involved with placental function, such as transforming growth factor-β (TGFβ), insulin-like growth factor (IGF), VEGF, and fibroblast growth factor (FGF) [30]. In a 2018 review, Giannakou et al. demonstrates that the plasminogen activator inhibitor-1 4G/5G(PAI-1 4G/5G) polymorphism gene, in particular, may be associated with the pathogenesis of preeclampsia in comparison to 26 other genetic variants that had associations with increased preeclampsia risk [31]. Another 2023 GWAS found that genes related to blood pressure traits are associated with preeclampsia and dysfunction of genes related to maintenance of successful pregnancy are associated with preeclamptic symptoms [32].

Immunologic factors are also thought to play a role in the pathogenesis of preeclampsia. In patients who did not have prior exposure to paternal or fetal antigens, like nulliparous patients, patients who changed partners between pregnancies, those with longer pregnancy intervals, patients who conceived in the first in vitro fertilization cycle, or those who conceived via intracytoplasmic sperm injection, there was a higher risk of preeclampsia [33]. There is also an immunologic role in preeclampsia thought to be similar to the pathophysiology involved in organ rejection, where natural killer (NK) cell receptors interact with HLA class I antigens on extravillous trophoblast (EVT) cells to regulate placental implantation [34,35]. The exact mechanism by which the interaction of NK cells and EVT cells plays a role in placental implantation is not known, but the increase in NK cell activity is thought to be a factor in placental implantation abnormalities [36,37]. The role of the immune system in preeclampsia has similarities to the role of immune cells in the pathophysiology of another vasoactive disease, hereditary angioedema [38].

Additionally, patients with preeclampsia were found to have decreased regulatory T cells, which are thought to protect the fetus from inflammatory responses, thereby increasing the risks of abnormal placentation [39]. While there is some evidence suggesting that patients with the KIR-AA genotype and fetal HLA-C2 genotype have a higher risk for preeclampsia, there are other studies that have found no definitive evidence that specific HLA alleles are associated with preeclampsia, leading to the general thought that the interaction between maternal, paternal, and fetal HLA types play a larger role in the immunologic pathophysiology of preeclampsia than that of the individual genotype [40,41].

## 5. Diagnostic Tests

### 5.1. Routine Testing

- Blood pressure readings (systolic pressure/diastolic pressure)
- Urine testing
  - Urine protein-to-creatinine ratio
  - Quantitative 24 h urine collection for total protein
  - Random urine protein measurement
  - Qualitative urine dipstick
- Complete blood count (CBC) with platelets
- Serum creatinine level
- Liver chemistries (aspartate aminotransferase [AST], alanine aminotransferase [ALT]), and bilirubin

### 5.2. Additional Testing under Certain Clinical Conditions

- Liver chemistries (including lactate dehydrogenase [LDH])
- Additional studies for those with liver dysfunction or epigastric or abdominal pain including glucose, amylase, lipase, and ammonia levels
- Coagulation studies (prothrombin time [PT], partial thromboplastin time [PTT], fibrinogen)
  - Additional ADAMTS13 studies in patients with thrombocytopenia (platelet count < 50,000/microL), fragmented blood cells on peripheral blood smears, neurologic findings, and normal clotting screen

### 5.3. Non-Routine Testing

- Serum plasma and urinary antiangiogenic markers for soluble fms-like tyrosine kinase-1 (sFlt-1), and their ratios
- Serum and urinary angiogenic markers for placental growth factor (PlGF), and their ratios

## 6. Testing Procedures

The procedures to diagnose preeclampsia include blood pressure and proteinuria assessment, laboratory assessment, and assessment of fetal status.

### 6.1. Blood Pressure Measurements

Elevated blood pressures after 20 weeks of gestation are diagnostic of hypertensive disease in pregnancy, including preeclampsia. Specifically, the definition of preeclampsia includes a systolic blood pressure measurement of $\geq$140 mmHg and/or diastolic blood pressure $\geq$ 90 mmHg at least two separate times, four hours apart [42]. Preeclampsia with severe features is defined as a systolic measurement $\geq$ 160 mmHg and/or diastolic blood pressure $\geq$ 110 mmHg on two occasions at least 4 h apart [2,43]. Blood pressure measurements occur at every obstetrical appointment for hypertensive disease screening. Blood pressure measurements must be performed appropriately or else considerable variability will be seen in its measurements [9,44].

The standard procedure to measure blood pressure involves the use of a sphygmomanometer and the auscultatory method. Although mercury sphygmomanometers are considered the gold standard, mercury has been phased out of clinical use due to its

potentially dangerous effects. Aneroid sphygmomanometers and digital oscillometric (automated) measuring devices are used more frequently in clinical settings [45,46]. Proper measurement of blood pressure depends on the time of measurement, proper calibration, the type of device used, appropriate cuff size and placement, patient position during the measurement, and the number of measurements. The standard location for placement of the blood pressure cuff is on the brachial artery on a bare arm [45].

Sources of error in blood pressure measurement can come from the effects of the patient's posture during the blood pressure reading. While most patients are sitting, there is not a general consensus on the best posture for a blood pressure measurement which could lead to differences in blood pressure data. Additionally, if a patient is engaged in certain activities prior to the measurement such as smoking or exercise, or is speaking during the measurement, that can result in an unrepresentative measurement. It is recommended that the patient be in the semi-reclining position with back support, have their arm at the level of the heart, and have their feet flat on the floor, with 5 min of rest prior to initiating the measurement. Patients should be advised to avoid caffeine and nicotine 30 min prior to blood pressure measurements, as they can elevate the measurement. The placement of the blood pressure cuff and the position of the arm can also change the readings. Lastly, some patients exhibit white coat hypertension, which causes their blood pressure to be consistently high in a clinical setting and normal outside of a clinical setting, which can also skew results [45].

### 6.2. Urine Protein Measurement

ACOG states that significant proteinuria during pregnancy can be diagnostic of preeclampsia and is specifically defined as a 24 h protein >300 mg or a protein-to-creatinine ratio of ≥0.3 protein/mg creatinine. The urine protein to urine creatinine ratio (UPCR) is the preferred method to analyze patient kidney function because of its convenience, reproducibility, and high accuracy [43,47,48]. While quantitative methods of measuring proteinuria are preferred, a qualitative urine dipstick of ≥2+ can also be highly suggestive of preeclampsia and can be useful as a primary screening method [49,50].

A quantitative proteinuria test to diagnose preeclampsia is a 24 h urine protein collection that begins when the patient wakes up and after discarding the first void, every subsequent void for 24 h is collected in a basin urinal, which is then transferred to a bottle. The last void collected should be 24 h from the first void, with no more than a 5–10 min variability from the time. The bottle of urine collection can be stored at room temperature for one to two days, but after two days needs to be kept cool and/or refrigerated. Twenty-four hour urine protein tests are found to be frequently inaccurate in pregnant patients due to over or under collection and the adequacy of collection should be called into question if it is outside of the range of 15 to 20 mg/kg of prepregnancy weight [51]. In fact, isolated proteinuria, using the widely-accepted threshold of a 24 h protein >300 mg or a protein-to-creatinine ratio of ≥0.3 protein/mg creatinine, can be found in 8% of pregnancies, while preeclampsia only occurs in 3–8% of pregnancies [52]. There is also evidence that proteinuria is increased in pregnant patients who are older than 35 years and overweight or obese [53].

The preferred UPCR is calculated from a random urine spot sample that is obtained from a clean catch. UPCR is defined as the urine protein concentration (in mg/dL) divided by the urine creatinine concentration (in mg/dL). This ratio can also be used to estimate total protein excretion over a 24 h period [54,55]. For random urine spot samples, it is recommended to use an AM measurement rather than a PM measurement due to greater median values more indicative of preeclampsia [56].

There are limited data on the relationship between increasing proteinuria and adverse outcomes of preeclampsia, which is why other clinical features are used to diagnose preeclampsia in the absence of proteinuria, such as visual symptoms, low platelet count, edema, and impaired liver or renal function. Additionally, it is a long-standing practice to measure proteinuria via a urine dipstick, which has been shown to have variable

and poor performance, especially in detecting preeclampsia in hypertensive pregnant women [9,43,57,58].

The standard procedure for a urine dipstick is to collect the urine sample mid-stream to avoid contamination from skin flora. In female patients, this should be performed prior to a pelvic exam to avoid contamination from vaginal secretions. The urine dipstick test strip should then be placed into the sample and blotted. It is important to wait 2 min for the reagents on the paper to fully react with the sample [59].

Urine dipstick test strips are semi-quantitative and usually have five squares that test for blood, glucose, protein, ketones, and pH. The protein square primarily detects albumin, and the results of the test can range from negative, indicating no urine protein, to +4, indicating >1000 mg/dL urine protein. Urine dipsticks have high false positive and false negative rates in pregnant patients due to the variability of urine osmolality during pregnancy. Although there is a low diagnostic accuracy for proteinuria in pregnancy, a score of 2+ or greater on the urine dipstick can be diagnostic of preeclampsia if quantitative urine analysis is unavailable [2].

*6.3. Serum Markers*

Routine lab work, including complete blood count (CBC) with platelets, liver chemistries, and serum creatinine, can be used to screen for preeclampsia. These samples are venous blood samples drawn from the median cubital vein in adults and collected in primary evacuated blood tubes. Venous blood sample collection preparation includes collection of the required materials, sanitization of the selected puncture area, and venous dilation, most often using a rubber tourniquet 5 to 10 cm proximal to the venipuncture site [60]. The tubes are chosen specifically for the test being ordered. For example, lavender-cap tubes are used for CBC, gold-cap tubes are used for liver chemistries, and non-routine labs, such as PlGF and sFlt, can be collected using red-cap or lavender-cap tubes, depending on the institution doing the analysis [61].

In a patient with preeclampsia, a CBC with platelets can show hemoconcentration, due to contraction of intravascular space and capillary leaking. While hematocrit values are usually increased in preeclampsia, if there is concurrent hemolysis, the hematocrit values may appear normal. Thrombocytopenia is the most common coagulation abnormality associated with preeclampsia, and a platelet count of <150,000/microL is seen in about 20% of patients with preeclampsia. While coagulation studies are not routinely tested in patients with preeclampsia, studies show that PT, PTT, and fibrinogen levels are not usually affected unless there is severe thrombocytopenia, liver dysfunction, bleeding, or placental abruption present [62,63].

Serum creatinine is routinely checked during pregnancy and, in normal pregnancies, there is a physiologic decrease in serum creatinine due to an increase in the glomerular filtration rate (GFR). In normal pregnancies, serum creatinine can decrease to a range of 0.4 to 0.8 mg/dL. In preeclampsia, serum creatinine is generally elevated, but it can also be measured in the physiologic 0.4 to 0.8 mg/dL range. A serum creatinine level >1.1 mg/dL is indicative of severe preeclampsia [64,65].

Serum liver chemistries, including AST, ALT, bilirubin, and LDH are increased in severe preeclampsia. Specifically, the elevation of AST and ALT to twice the upper limit of normal is seen in severe forms of preeclampsia. Elevation of bilirubin specifically indicates hemolysis, and elevation of LDH can be evidence of liver dysfunction, hemolysis, or both. While AST, ALT, and bilirubin are checked routinely in pregnancy, LDH and serum levels of glucose, amylase, lipase, and ammonia are reserved for pregnant patients with known liver dysfunction or abdominal pain [66,67].

Non-routine serum testing for preeclampsia includes testing for angiogenic markers like sFlt-1 and PlGF. A randomized control trial conducted in 2019 utilizing sFlt-1 and PlGF for patients with potential preeclampsia, showed that the sFlt-1/PlGF ratio, when used with standard clinical practices for diagnosis of preeclampsia, improved clinical precision in diagnosing preeclampsia without changes in the admission rate. There was also a

trend noted toward more rapid diagnosis of preeclampsia in the research arm that used sFlt-1 and PlGF ratios, but this trend was statistically not significant [68]. A 2023 review reported that measurement of sFlt-1/PlGF ratios can differentiate between mild and severe forms of preeclampsia as well as between gestational hypertension and preeclampsia, with higher ratios associated with more severe disease [69]. The sFlt-1/PlGF ratio has also been shown to have similar progression patterns in preeclampsia and a preeclampsia-associated syndrome involving hemolysis, elevated liver enzymes, and a low platelet count (HELLP syndrome) [70,71].

These angiogenic factors have been noticed to vary based on early- versus late-onset preeclampsia, but their clinical value in differentiating between early-onset and late-onset preeclampsia has varied results. PlGF levels have been shown to be lower in patients with early-onset preeclampsia than in those with late-onset [5]. The changes in VEGF levels between early-onset and late-onset preeclampsia has not been well studied, but preliminary research has found no significant difference [5,72]. While levels of sFlt-1 are increased in preeclampsia overall, they have been found to be higher in early-onset preeclampsia compared to late-onset [5].

The combination of using sFlt-1 and PlGF markers for preeclampsia diagnosis has high sensitivity and specificity for prediction of early onset preeclampsia [72]. The sFlt-1/PlGF ratio has been used to show that preeclampsia is a continuously progressive disease, resulting in an anti-angiogenic state [71]. Stepan et al. outline the use of sFlt-1/PlGF ratio based on gestational age to aid in diagnosis of preeclampsia. As per their proposed guidelines, a sFlt-1/PlGF ratio > 85 would signify early-onset preeclampsia and a ratio of >110 would indicate late-onset preeclampsia [73]. Recent studies demonstrate that an sFlt-1/PlGF ratio of 38 or lower could be used to predict the short-term absence of preeclampsia and that the sFlt-1/PlGF ratio is also effective at predicting preterm delivery in both early-onset and late-onset preeclampsia [69,74]. More research is needed regarding the trends and utility of using these angiogenic factors to differentiate between early-onset versus late-onset preeclampsia.

Unfortunately, as these serum markers are not routinely collected, there is limited evidence regarding using these markers as a screening tool, and more evidence is needed to determine the significance of using angiogenic serum markers in the screening and diagnosis of preeclampsia [9,19].

There are several factors that can lead to sources of error when analyzing venous blood samples, including collecting in the wrong container, contamination of anticoagulants, or additives from other collection tubes. It is important to ensure that the proper tube is being used for the desired analysis [61,75].

*6.4. Ultrasound and Doppler Studies*

Assessment of fetal status can occur with the use of an ultrasound. When preeclampsia clinically develops before term, fetal growth restriction can be seen due to reduced uteroplacental perfusion. When preeclampsia develops at term, fetal growth is usually preserved [24].

Doppler studies can be used to estimate the flow of blood in uterine arteries and in uteroplacental maldevelopment, there is increased uterine artery notching and pulsatility, which can be seen on Doppler scans. While uteroplacental maldevelopment can be associated with preeclampsia, uterine artery pulsatility, and notching are not specific nor sensitive to preeclampsia and are not used for diagnosis [76,77].

Overall, there are limited data regarding screening tools for preeclampsia beyond blood pressure measurements and proteinuria, due to a greater research focus on the etiology and treatment of preeclampsia [24].

## 7. Clinical Significance

Preeclampsia is a complex disease with multisystem effects that affect both the mother and fetus during pregnancy. Patients with preeclampsia present with hypertension and can

also have associated symptoms, including persistent/severe headache, visual abnormalities, altered mental status, abdominal pain, epigastric pain, dyspnea, or orthopnea. As a result, patients with preeclampsia have an increased risk for cerebrovascular hemorrhage, pulmonary edema, acute kidney injury, liver rupture, abruption, and eclampsia [78,79].

The headache experienced with severe preeclampsia is usually throbbing or pounding and persists despite the use of analgesics. While the exact mechanism for severe headaches during preeclampsia is not fully understood, it is thought that dysregulation of cerebrovascular autoregulation leads to sporadic vasodilation and vasoconstriction [80].

Visual symptoms experienced during preeclampsia include blurred vision, flashing lights/sparks (photopsia), dark areas in the visual field (scotomata), diplopia, or blindness. Similar to the proposed pathophysiology for headaches in preeclampsia, visual changes are also thought to occur due to cerebrovascular dysregulation and retinal arteriolar spasm [81].

The neurologic impairment seen in preeclampsia, such as increased brain water content and the loss of cerebrovascular regulation, is associated with cognitive impairment, depression, anxiety, executive dysfunction, and post-traumatic stress disorder. Patients who exhibit these outcomes are also associated with low fetal birth weight, intrauterine growth restriction, and premature birth [82,83].

The abdominal pain experienced during preeclampsia can be attributed to liver tenderness due to stretching of Glisson's capsule due to hepatic bleeding or swelling. While liver hemorrhage is rare, it should be considered on the differential when there is a sudden onset of right upper quadrant pain with an associated decrease in blood pressure [84]. A dangerous complication associated with preeclampsia can also present with abdominal pain, and it involves hemolysis, elevated liver enzymes, and a low platelet count (HELLP syndrome) due to an exaggerated inflammatory and oxidative stress response [70]. In a few cases, abdominal pain in HELLP syndrome can be due to hepatic rupture, which leads to adverse outcomes, increasing both maternal and fetal morbidity and mortality [85–87]. Epigastric pain occurring during preeclampsia can be indicative of acute pancreatitis, which is a rare complication of preeclampsia [88].

Pulmonary edema in preeclampsia can present with shortness of breath, cough, wheezing, or chest pain. A combination of these symptoms with a decreased oxygen saturation of <93% is a predictor for adverse maternal outcomes. The pathophysiology for pulmonary edema, and edema in general, during preeclampsia, is thought to occur from increased vascular hydrostatic pressure and decreased plasma oncotic pressure, but other reasons for pulmonary edema in preeclampsia include capillary leaking due to endothelial activation, left heart failure, and volume overload. There is also considerable overlap seen between preeclampsia and peripartum cardiomyopathy [89,90].

Stroke is the most debilitating complication of severe preeclampsia. Most preeclampsia-associated strokes are hemorrhagic, and reducing blood pressure reduces the risk of stroke. Strokes in pregnant patients can present as headache, seizure, dizziness, nausea, and visual symptoms [91]. Data from the Nationwide Inpatient Sample (NIS) show that, of pregnancy-associated strokes, 96% of the patients had a pre-stroke systolic blood pressure reading ≥160 mm Hg, indicating that preeclampsia is a risk factor for stroke [92,93].

Seizure in patients with preeclampsia advances the diagnosis to eclampsia. During an eclamptic seizure, fetal bradycardia is present, which usually becomes fetal tachycardia upon resolution of the seizure. In some cases, the fetal heart rate can become nonreassuring, prompting urgent resuscitation. Eclamptic pregnancies have a higher risk of preterm birth, placental abruption, and intrauterine asphyxia, which can all lead to perinatal death [94,95].

Oliguria in preeclampsia can occur in labor or within 24 h of the postpartum period. This occurs due to vasospasms decreasing intravascular space, causing sodium and water retention, thereby resulting in intrarenal vasospasm. As a result, GFR can fall by 25%, and urine output may be <500 mL/24 h, leading to kidney injury [96].

Placental abruption is a dangerous phenomenon that can happen during pregnancy for both the mother and fetus and occurs in about 1% of patients with preeclampsia without severe features, and 3% in patients with preeclampsia with severe features [97].

In patients with preeclampsia, their fetuses can experience fetal growth restriction and possibly preterm birth, which can increase perinatal mortality [3,98,99].

## 8. Discussion

Preeclampsia is a multisystem disease that affects 2–10% of pregnancies around the world. While the exact etiology and pathophysiology of preeclampsia is not fully understood, it is thought to involve abnormal placental implantation, vascular endothelial damage and oxidative stress, anti- and pro-angiogenic factors, genetics, and immunologic factors. Current routine screening guidelines for preeclampsia include blood pressure measurement and urine analysis. Studies have shown that serum markers of anti-angiogenic factors, such as sFlt-1, and pro-angiogenic factors like PlGF and VEGF when measured in relation to each other, are correlated to adverse preeclamptic outcomes. However, due to a greater focus on the treatment and prevention of preeclampsia, these serum markers are not yet included in routine screening. More evidence is needed regarding using serum angiogenic biomarkers as a screening tool for preeclampsia to expedite the diagnosis and eventual treatment for patients with preeclampsia and subsequently reduce morbidity and mortality.

Preeclampsia is a complex disease affecting both the patient and fetus, and to effectively screen for it, there must be a team of healthcare workers dedicated to providing the best care for pregnant patients. Additionally, patient education regarding preeclampsia risk factors is of utmost importance as early diagnosis of preeclampsia can reduce adverse pregnancy outcomes. Future directions include a deeper understanding of the incorporation of standard screening tools, such as blood pressure measurement and urine analysis, with newer laboratory screening markers to optimize and prevent disease with better maternal and fetal outcomes.

**Author Contributions:** A.F.O. and M.J. wrote and read all the manuscript versions for intellectual content. All authors have read and agreed to the published version of the manuscript.

**Funding:** This research received no external funding.

**Institutional Review Board Statement:** Not applicable.

**Informed Consent Statement:** Not applicable.

**Data Availability Statement:** No new data were created or analyzed in this study. Data sharing is not applicable to this article.

**Conflicts of Interest:** The authors declare no conflicts of interest.

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
