# Peer review of "Review of Laboratory Testing and Biomarker Screening for Preeclampsia"

_2673-8430, doi:10.3390/biomed4020010_

Round 1
Reviewer 1 Report
Comments and Suggestions for Authors
In the present manuscript, the authors explore various laboratory and biomarker tests for preeclampsia screening and diagnosis. Routine tests include blood pressure readings, urine tests, platelet counts, serum creatinine, liver chemistries, and serum bilirubin levels. Non-routine tests include angiogenic and anti-angiogenic markers like PlGF and sFlt-1. Blood pressure measurements and proteinuria are the most commonly used screening tools.
In the manuscript, there are several Major Issues that should be addressed in order to improve value of this paper.
1. First of all, the paragraph with the bibliography is missing, so it is impossible to evaluate the cited references. The authors should correct this shortcoming. Anyway, I suggest citing the publication “Ferrara AL et al, doi: 10.1007/s12016-021-08842-9”, in which the release of vasoactive mediators by immune cells is discussed in depth.
2. Authors should consider introducing a paragraph describing the search strategy.
What are the inclusion/exclusion criteria for the papers discussed in the study? How were the articles searched for? Which keywords were chosen? Which database was used? What date was the search carried out? How many papers were included in the review in total?
3. In the paragraph “3. Pathophysiology”: Authors should consider inserting a Table 1 since it is difficult to follow the paragraph, and dividing it as below:
Table 1. Release of vasoactive factors in the pathophysiology of preeclampsia.
|
Vasoactive factor |
Expression |
Pathophysiological condition |
Cellular source |
Effects |
Testing Procedures |
Years |
Ref. |
|
VEGF |
Up |
Hypoperfusion Ischemia Hypoxia |
Differentiation of cytotrophoblasts |
Maternal vascular injury Inflammation |
Proteinuria ….. |
----- |
[n°reference]
|
|
|
|
|
- |
||||
Abbreviation: VEGF, …..; ns, not specified (where it is not defined in the cited paper).
Minor Issues:
1. Check the unit of measurement
2. Check the spaces between words and references
3. Check the punctuation
4. Check the abbreviations
Reviewer 2 Report
Comments and Suggestions for Authors
The manuscript attempted to touch very important topic, the text is generally well structured and written. However, it would benefit from the following points:
- The paper lacks one significant issue – how the biomarkers described and analyzed in the paper are relevant to two types of PE, early-onset and late-onset PE. Are there any differences between expression of various biomarkers upon development of these PE forms? Such information should be added to the text.
- Next, please, discuss whether the diagnosis of early-onset PE is even possible.
- Also discuss how to distinguish between PE and other pregnancy-associated hypertensive disorders using the biomarkers analyzed in the manuscript.
- There is no references list in the text!
Round 2
Reviewer 1 Report
Comments and Suggestions for Authors
The authors did an excellent job implementing the necessary adjustments.
Author Response
Thank you very much for taking the time to review this manuscript.
Reviewer 2 Report
Comments and Suggestions for Authors
Table 1: PlGF is named platelet-derived growth factor! Please, carefully check an entire text to search for similar errors.
